# Effect of frequency and rhythmicity on flicker light-induced hallucinatory phenomena

Ioanna Alicia Amaya[1,2,3], Nele Behrens[1,4], David John Schwartzman[5], Trevor Hewitt[5], Timo Torsten Schmidt[1] *

1 Department of Education and Psychology, Freie Universität Berlin, Berlin, Germany, 2 Charité – Universitätsmedizin Berlin, Einstein Center for Neurosciences Berlin, Berlin, Germany, 3 Berlin School of Mind and Brain, Humboldt-Universität zu Berlin, Berlin, Germany, 4 Department of Psychology, Sigmund Freud University Berlin, Berlin, Germany, 5 Sackler Centre for Consciousness Science and Department of Informatics, University of Sussex, Brighton, United Kingdom

* timo.t.schmidt@fu-berlin.de

**Data Availability Statement:** All questionnaire data files are available on the OSF data repository (URL: https://osf.io/5d29g/).

## Abstract

Flicker light stimulation (FLS) uses stroboscopic light on closed eyes to induce transient visual hallucinatory phenomena, such as the perception of geometric patterns, motion, and colours. It remains an open question where the neural correlates of these hallucinatory experiences emerge along the visual pathway. To allow future testing of suggested underlying mechanisms (e.g., changes in functional connectivity, neural entrainment), we sought to systematically characterise the effects of frequency (3 Hz, 8 Hz, 10 Hz and 18 Hz) and rhythmicity (rhythmic and arrhythmic conditions) on flicker-induced subjective experiences. Using a novel questionnaire, we found that flicker frequency and rhythmicity significantly influenced the degree to which participants experienced simple visual hallucinations, particularly the perception of Klüver forms and dynamics (e.g., motion). Participants reported their experience of geometric patterns and dynamics was at highest intensity during 10 Hz rhythmic stimulation. Further, we found that frequency-matched arrhythmic FLS strongly reduced these subjective effects compared to equivalent rhythmic stimulation. Together, these results provide evidence that flicker rhythmicity critically contributes to the effects of FLS beyond the effects of frequency alone, indicating that neural entrainment may drive the induced phenomenal experience.

## Introduction

Flicker light stimulation (FLS) reliably induces simple visual hallucinations in healthy participants via closed-eye ocular stimulation with stroboscopic light [1–3]. Simple visual hallucinations, synonymous with elementary visual hallucinations, refer to the subjective experience of colours and geometric patterns that are devoid of semantic content. The experience is often accompanied by other phenomenological changes, such as altered mood, arousal, and sense of time passing [2]. Simple visual hallucinations experienced under FLS display marked similarities to the perceptual changes associated with migraine aura [4,5], epileptic seizures [6] and Charles Bonnet Syndrome (hallucinatory experiences due to sensory deprivation resulting from macular degeneration) [7] as well as drug-induced psychedelic experiences [2,8]. In

**Funding:** The investigator-initiated study was financially supported by a donation from Lumenate Growth Ltd to Freie Universität Berlin allocated to TTS. There was no additional external funding received for this study.

**Competing interests:** I have read the journal's policy and the authors of this manuscript have the following competing interests: TTS: This research was supported by an unrestricted donation from Lumenate Growth ltd to Freie Universität Berlin allocated to TTS. This does not alter our adherence to PLOS ONE policies on sharing data and materials.

recent years, FLS has been used as an experimental tool to study the neural underpinnings of visual hallucinations [9]. However, in order to draw links between neural mechanisms underlying specific forms of visual hallucinations, it is first important to establish a thorough characterisation of the experienced phenomena.

Flicker-induced effects were first formally described by Purkinje in 1819 [10]; thereafter, the phenomenon was relatively unexplored until the invention of the electroencephalogram (EEG) in the 1920s [11]. This allowed the observation of synchronised brain oscillations when FLS was presented in the alpha frequency range (8–12 Hz) [12]. Later, it was explored recreationally in the 1960s with the creation of the "Dreamachine", a low-fi method of delivering FLS using a record player [11]. Today, FLS can be delivered using specially programmed electronic lamps with precise manipulation of flicker frequency, rhythmicity (i.e., the temporal pattern of flashes), and brightness.

Initial studies attempting to characterise visual experiences arising from altered and pathological states employed illustrations and open report methods, which revealed a striking universality in the types of visual patterns experienced [13,14]. Four of these commonly occurring patterns are collectively named the Klüver form constants [15], which are comprised of grids, spirals, tunnels, and targets (see [16,17] for illustrations). FLS-induced perception of motion and colours are also shared across various pathologies and altered states. For example, migraine sufferers often report seeing red, yellow, and blue in addition to bright white [18] and vivid colours are one of the most frequently reported characteristics of N, N-dimethyltryptamine (N, N-DMT) experiences [19]. In addition, FLS also induces other types of imagery that have lower levels of pattern organisation and higher degrees of noise, such as TV static and floating, scattered blobs and dots. These are sometimes referred to as phosphene forms and likely occur due to retinal stimulation with a strong light source [20]. Aside from simple hallucinations, complex visual hallucinations (i.e., realistic scenes, objects, and faces) have also been reported during FLS, albeit less frequently [2,3]. Recent research further explored if FLS experiences relate to person-specific factors, such as the personality trait of Absorption [2,21]. It was also found that people with Aphantasia (i.e., lack of mentally simulated visual imagery [22]) report fewer FLS-induced visual effects [23].

Recent studies have used standardised methods to assess the flicker-induced subjective experience, e.g., Bartossek *et al.* [2] administered the Altered States of Consciousness Rating Scale (5D-ASC/11-ASC; [24]) and the Phenomenology of Consciousness Inventory (PCI) [25]. These questionnaires are well-established and validated to assess a whole spectrum of altered experiences and thereby allow comparisons across different types of altered states of consciousness [8]. However, due to their breadth of measured phenomena, they are limited in capturing a high level of detail of visual effects that would enable differentiation between different types of visual hallucinations. Using an analogue slider to assess experience intensity, Schwartzman *et al.* [3] were able to differentiate the intensity of experiences between different frequencies of FLS, which was otherwise not captured via 5D-ASC ratings. These observations highlight the need for a careful and detailed assessment of the types of visual phenomena experienced during FLS that extends beyond the currently available tools.

The link between phenomenology and neurophysiology can be used to shed light on the neural mechanisms underlying FLS-induced visual hallucinations. Using periodic flicker (i.e., FLS with regular inter-flash intervals; also called rhythmic flicker), it was found that FLS at alpha frequency (8–12 Hz) induces stronger simple visual hallucinations than other frequencies [2,3,17] and additionally enhances the amplitude of EEG oscillations at the targeted frequency band of stimulation [26,27]. This indicates that entrainment (i.e., synchronisation of brain oscillations with periodic external driving stimulation) may contribute to the generation of simple visual hallucinations. To further test whether entrainment is indeed a driving factor

in generating the subjective experience associated with FLS, one must compare the subjective effects between traditional rhythmic stimulation and frequency-matched arrhythmic stimulation. Theoretically, removing the rhythmicity of the stimulation should abolish entrainment. Therefore, if arrhythmic frequency-matched FLS produces fewer subjective effects, it would indicate that neural entrainment contributes to the generation of FLS-induced simple visual hallucinations.

Here, we aim to determine the effects of frequency and rhythmicity on flicker-induced phenomenology. Based on previous findings, we expect that rhythmic FLS within the alpha frequency range (8–12 Hz) will lead to increased reports of simple visual hallucinations, Klüver forms and visual experiences that are more dynamic (i.e., moving patterns, patterns changing frequently over time) and visually detailed, compared to other frequencies. We further expect that rhythmic FLS will generate more simple visual hallucinations compared to frequency-matched arrhythmic stimulation. We utilise two arrhythmic conditions that vary in their degree of arrhythmicity. We hypothesise that higher variability in inter-flash intervals (i.e., greater arrhythmicity) will lead to a greater reduction in subjective effects. In contrast, as all FLS conditions deliver the same total duration of light stimulation, we hypothesise that seeing phosphene forms (e.g., blobs, TV static) will be frequency- and rhythmicity-independent. This is because they have low levels of pattern organisation, making it likely they are caused by retinal stimulation and not higher-order neural mechanisms. We will also explore whether frequency and rhythmicity affect reports of complex visual hallucinations and types of colours that are observed during the flicker experience.

## Methods

### Participants

Healthy participants were recruited (*N* = 20; 12 female, 7 male, 1 diverse; age range 20–37 years, *M* = 24.78, *SD* = 4.25) that met the following inclusion criteria, as established by Bartossek *et al.* [2]: no history of epilepsy, migraines, psychological problems (e.g., depression, anxiety disorders), no current consumption of any psychotropic drugs (e.g., antidepressants, neuroleptics). To mitigate the risk of an adverse reaction, we only included subjects who had previously used FLS for recreational purposes. Alternatively, an EEG examination was performed to screen for indicators of photosensitive epilepsy, which would lead to exclusion. The recruitment took place via student mailing lists and through word-of-mouth. Participants gave their written consent before commencing the experiment. All materials and procedures were approved by the ethics committee at Freie Universität Berlin (application reference: 045/2021). Seventeen participants filled out the questionnaires in German, of which fifteen were native speakers, while the remaining three participants preferred English, two of which were native speakers.

### Materials

**Flicker light stimulation.**   A custom stroboscope was constructed by Lumenate Growth Inc. (Bristol, United Kingdom) to generate light stimulation. It consists of twelve 4500k J2 6V white LEDs organised in a three-by-four grid with dimensions 128 x 176mm (width x height). The lamp was set to deliver 5,520 Lumens over participants' eyes (maximum capacity is 10,360 Lumens). Positioned approximately 150cm from participants, the LEDs were within a visual angle of approximately 6.5˚, while the setup assured that illumination of the visual field was experienced as homogenous. The lamp was interfaced with an Arduino (v1.8.16) to deliver FLS at different frequencies and rhythmicities. Three rhythmicity conditions were used across four levels of frequency (3 Hz, 8 Hz, 10 Hz and 18 Hz) [Fig 1A]. Rhythmicity levels were:

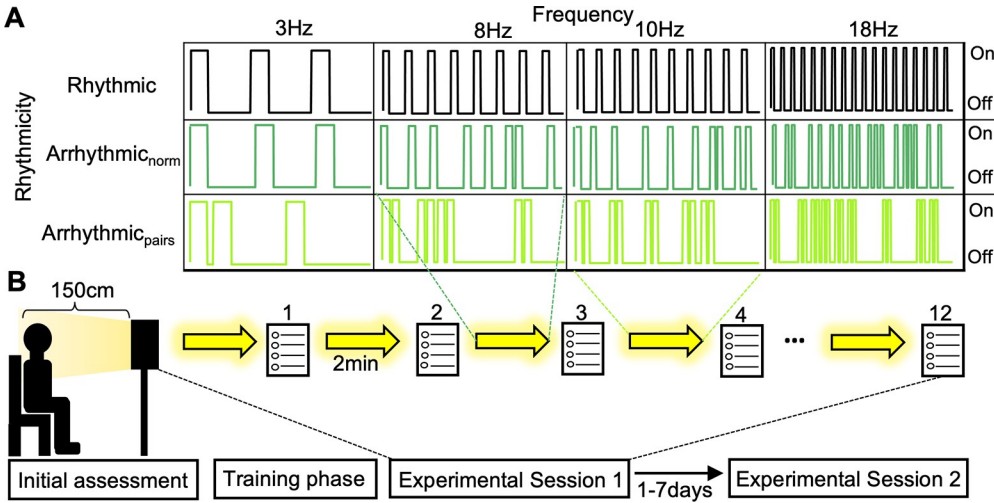

**Fig 1.** (A) The experiment comprised a 3x4 factorial design with 3 levels of rhythmicity (Rhythmic, Arrhythmic$_{norm}$ and Arrhythmic$_{pairs}$) and 4 levels of frequency (3 Hz, 8 Hz, 10 Hz, 18 Hz). (B) In the experimental setup, participants were seated in a dark room 150 cm away from the stroboscope (Lumenate Growth Inc., Bristol, United Kingdom). The initial assessment involved the completion of TAS questionnaire. In the training phase, participants were exposed to FLS and familiarised with the Stroboscopic Visual Experience Survey (SVES) and ASC-R items. The subsequent flicker session consisted of twelve two-minute stimulation periods presented in a fully randomised order of conditions. Following each stimulation period, participants rated their experience using SVES and ASC-R items. A second session took place within a week of the first.

Rhythmic, which consists of periodic light stimulation following a 0.3 duty cycle (30% ON time); normally distributed arrhythmic stimulation (Arrhythmic$_{norm}$), where inter-flash intervals (IFIs) were sampled from a normal distribution with mean IFI equal to one OFF time during periodic stimulation at matched frequency (e.g., 70 ms at 10 Hz) and standard deviation equal to 0.45*OFF time at matched frequency; and paired arrhythmic stimulation (Arrhythmic$_{pairs}$), which involved paired high frequency flashes (similar to the luminance control condition used by Ffytche [28]). The inter-pair OFF time was calculated as 100/freq ms and adjusted to 10 ms if the value would otherwise be lower. The flash pairs were embedded within a set of intervals that were sampled from an exponential probability distribution, where the mean IFI was equal to one OFF time at frequency-matched periodic stimulation. As IFIs were calculated for every second, 3 Hz Arrhythmic$_{pairs}$ used one pair and one single flash for every second. All conditions delivered 300 ms of light stimulation per second (30% ON time). While previous studies used 50% ON time [2,3], we decided to shorten the ON period as this allowed for a greater degree of variation in the arrhythmic IFIs. The Arrhythmic$_{pairs}$ condition contains IFIs with higher variability than the Arrhythmic$_{norm}$ at each frequency level (excluding the constant inter-pair interval). This was determined using the root mean squared of successive differences (RMSSD), which is commonly used for calculating heart rate variability [29] (e.g., at 10Hz, Arrhythmic$_{pairs}$: RMSSD = 166.57, Arrhythmic$_{norm}$: 64.22). For an illustration of the ON/OFF flicker sequences see Fig 1A.

## Questionnaires

Three questionnaires were used in the study: The Tellegen Absorption scale (TAS), selected items from the Altered States of Consciousness Rating Scale (ASC-R) and an abridged version of the novel Stroboscopic Visual Experience Survey (SVES). Participants were able to answer the questionnaires in English or German.

**TAS.** TAS assesses the personality trait Absorptiveness. It captures the openness to experiencing mind-altering states with 34 items rated on a five-point Likert scale (0 = "Not at all" to 4 = "Very much") [21].

**ASC-R.** Eleven questions were taken from the Altered State of Consciousness Scale [24]. The ASC-R is a standardised and validated research tool widely used to investigate ASCs [8,24]. Items are rated using a visual analogue scale (VAS) ranging from "no, not more than usually" to "yes, much more than usually". The 96-item questionnaire can be decomposed into 11 factors, including subscales from the main five dimensions: Oceanic Boundlessness, Visionary Reconstruction, Dread of Ego Dissolution, Auditory Alterations, Vigilance Reduction) [30]. As FLS primarily induces visual effects, we selected all items from the Elementary Imagery subscale: ("I saw regular patterns [with closed eyes or in complete darkness.]"; "I saw colors [with closed eyes or in complete darkness.]"), excluding the item "I saw brightness or flashes of light with eyes closed or in complete darkness" as the experience is inherent to FLS. Further, we selected all items from the Complex Imagery subscale of the Visionary Reconstruction scale: "I saw whole scenes roll by [with closed eyes or in complete darkness]"; "I could see images from my memory or imagination with extreme clarity"; "My imagination was extremely vivid". As it has been reported that other altered state phenomena can additionally arise from FLS [2], we also selected two items from the Positive Derealisation subscale: "I felt as if in a wonderful other world."; "The boundaries between myself and my surroundings seemed to blur", the item "My sense of space and time was altered as if I was dreaming" from Altered Perception of Time subscale and "I had the impression I was out of my body." from the Positive Depersonalisation subscale of the Oceanic Boundlessness scale. Finally, to measure participants alertness during each trial we included "I felt sleepy" from the Reduction of Vigilance scale.

**SVES.** An abridged version of the Stroboscopic Visual Experience Survey, which is currently under development, was used. The SVES is a computer-based questionnaire designed to allow participants to capture aspects of their FLS experience more accurately, implemented using the SoSci Survey platform. It was originally constructed in English and was translated into German for the purposes of this study. The SVES begins with an instruction page that explains how to answer each item. Thereafter, participants are asked "How well do you recall your visual experience right now" and "What colours did you see? Select all that apply". There are twenty-four colour options, which correspond to the following Natural Colour System (NCS) IDs: S1040-R, S2070 Y80R, S4050 Y90R, S2050-Y50R, S1070-Y70R, S3050-Y80R, S0550-Y20R, S0580-Y30R, S1060-Y40R, S2070-G70R, S2070-G60Y, S5040-B70G, S0520-B, S3050-R70B, S4050-B10G, S1020-R50B, S5020-R70B, S4050-R50B, S2050-R20B, S5010-B70G, S6010-G10Y, S4050-R20B as well as black and white. Participants are then asked to rate the occurrence of different patterns and forms during the preceding FLS experience (See S1 Appendix for full list of items). The patterns used in these questions are based on geometric patterns that were reported in previous FLS studies [1,16,28] and additional piloting. The patterns vary between Klüver form constants, phosphenes forms and other possible geometric patterns that could appear, as well as one geometric pattern that is unlikely to occur (akin to a control pattern). In addition, overarching visual aspects of the FLS experience are assessed with another ten items, such as "Did your visual experience continuously change or evolve over time?" and "Did your visual experience contain a high level of randomness or chaos?". These items use a visual analogue scale (VAS) ranging from 0 (no, not at all) to 100 (yes, very much), which was used to increase comparability of effect sizes with the ASC-R. Three example pictures are given to demonstrate the range of possibilities across the scale. Item 1 was excluded due to technical difficulties. Item 16 and Item 22 were excluded due to high response variability. For the subsequent analysis, items were grouped together that conceptually measured the same visual phenomena. Items were grouped into the following scales: colors (Item

2), simple visual hallucinations (Item 3—Item 10), phosphene forms (Item 11 & Item 12), detail (Item 13), dynamics (Item 14 & Item 15), paisley (Item 17), complex visual hallucinations (Item 18) and absorption (Item 21). The paisley pattern represents a geometrically simple pattern that is unlikely to occur. Two subscales of the simple visual hallucinations scale were also determined: Klüver (Item 3 –Item 6) and other (Item 8 and Item 10) in order to assess whether there were differences in reported pattern subtypes.

## Experimental procedure

**Initial assessment.** A semi-structured interview, which followed the guidelines published by Bartossek *et al.* [2], took place to screen participants for eligibility. During the initial assessment [Fig 1B], participants were given an information sheet and then filled out the consent form. A pseudo-anonymised subject ID was created to link data from the two experimental sessions. Participants completed the TAS via tablet.

**Training phase.** Participants wore noise-cancelling headphones and were seated on a comfortable chair with headrest 150 cm away from the lamp in a dark room. The training phase consisted of four one-minute stimulation periods: constant light, 3 Hz, 10 Hz and 18 Hz of rhythmic flicker light. This allowed participants to accustomise to the light intensity and type of experience. Next, participants were asked to evaluate a static image using the SVES to gain familiarity with the questionnaire items.

**Experimental sessions.** If participants had no further questions, the experimental phase could begin. This involved presentation of twelve two-minute stimulation periods with a fully randomised order of conditions. The conditions were comprised of three levels of rhythmicity (Rhythmic, Arrhythmic$_{norm}$ and Arrhythmic$_{pairs}$) and four levels of frequency (3 Hz, 8 Hz, 10 Hz, 18 Hz). Following each stimulation period, participants answered the SVES and ASC-R items to evaluate their phenomenal experience. A second experimental session took place at the same time of day 1–7 days after the first.

## Statistical analysis

All statistical analysis was conducted using Rstudio (v1.4.1103). To test whether participant ratings differed across the two test sessions, 3x4x2 ANOVAs with rhythmicity, frequency and test session as factors were run. For each participant, the mean rating of each scale between the two sessions was used for further analysis. To test the effects of rhythmicity and frequency, we ran 3x4 ANOVAs with rhythmicity and frequency as factors. Post-hoc Tukey HSD-Tests were used to compare the distribution of ratings in different conditions. As there was insufficient evidence to assume normality of data for some of the assessed scales, shown with Shapiro-Wilks normality tests, we used Kruskal-Wallis tests to confirm the ANOVA results with non-parametric testing. To test the effect of rhythmicity and frequency on colour selection, we ran a repeated measures logistic regression model for each colour using the *lme4* package in R. In the regression model, Participant ID was included as a random effect term while frequency and rhythmicity were fixed effect terms. Further, we used Pearson product-moment correlation to explore whether there were associations between the personality trait Absorptiveness and the occurrence of complex visual hallucinations.

## Results

### Effects of test session order on flicker-induced phenomena

To test for the effects of test session order, we performed a 3x4x2 ANOVA for each SVES scale with frequency, rhythmicity, and test session as factors. We found effects of test session for the

*Simple Visual Hallucination* (F(1, 456) = 11.93, $p < 0.001$, $\eta^2_p = 0.03$), *Dynamics* (F(1, 456) = 16.41, $p < 0.001$ $\eta^2_p = 0.03$) and *Paisley* SVES scales (F(1, 456) = 3.87, $p = 0.05$, $\eta^2_p < 0.01$), albeit with small effect sizes, where ratings were higher in the first session compared to the second. When using nonparametric Wilcoxon Rank Sum tests to assess differences in ratings for each condition, there was no significant differences between test sessions for any SVES scale. Therefore, for subsequent analysis, the mean score was calculated from the first and second session for each participant.

## Effect of frequency and rhythmicity on simple visual hallucinations

We sought to test if frequency and rhythmicity affects how participants rated, between 0 and 100, the occurrence of simple visual hallucinations in their flicker experience. First, we ran a 3x4 ANOVA on the *Simple Visual Hallucination* scale of the SVES. We found main effects of frequency (F(3, 228) = 46.19, $p < 0.001$, $\eta^2_p = .38$) and rhythmicity (F(2, 228) = 27.91, $p < 0.001$, $\eta^2_p = .19$) and a significant interaction effect (F(6, 228) = 3.78, $p = 0.001$, $\eta^2_p = 0.09$). Nonparametric Kruskal-Wallis testing confirmed a significant effect of frequency and rhythmicity (H(3) = 79.3, $p < 0.001$; H(2) = 26.5, $p < 0.001$). Post-hoc Tukey tests found that 8 Hz, 10 Hz and 18 Hz stimulation elicited higher ratings of simple visual hallucinations than 3 Hz (all $p < 0.001$). Furthermore, all rhythmicity levels were significantly different from each other, where rhythmic was higher than Arrhythmic$_{norm}$ ($p = 0.001$) and Arrhythmic$_{pairs}$ ($p < 0.001$) and Arrhythmic$_{norm}$ was higher than Arrhythmic$_{pairs}$ ($p < 0.001$). For interaction effects, post-hoc Tukey tests showed that the highest increase in reports of simple visual hallucination was between Rhythmic and Arrhythmic$_{pairs}$ stimulation at 10Hz ($p < .001$) and remained significant for 18Hz ($p = .003$) and 8Hz ($p = .01$) [See S1 Table for full report of Tukey tests]. Furthermore, during Rhythmic stimulation, reports of simple visual hallucinations are significantly higher during 8Hz, 10Hz and 18Hz compared to 3Hz (all $p < .001$) [See S1 Table for full report of Tukey tests]. Fig 2A summarises the interaction between rhythmicity and frequency on ratings of simple visual hallucinations. Secondly, we tested for effects on simple visual hallucinations via scores of the ASC-R *Elementary Imagery* scale. Here, a 3x4 ANOVA revealed a main effect of frequency (F(3, 228) = 5.89, $p < 0.001$, $\eta^2_p = 0.07$), which was further confirmed by Kruskal-Wallis testing (H(3) = 18.6, $p < 0.001$). Again, post-hoc Tukey tests found that 8 Hz, 10 Hz and 18 Hz stimulation generated higher ratings of simple visual hallucinations than 3 Hz (all $p < 0.01$).

## Effects of frequency and rhythmicity on dynamics and detail

To test if frequency and rhythmicity would affect the visual dynamics and detail of hallucinatory phenomena, we ran 3x4 ANOVAs on the *Dynamics* and *Detail* SVES scales. We found a main effect of rhythmicity ($F$(2, 228) = 33.83, $p < 0.001$, $\eta^2_p = .23$) and frequency ($F$(3, 228) = 46.32, $p < 0.001$, $\eta^2_p = .38$) as well as an interaction effect ($F$(6, 228) = 3.50, $p = .0025$, $\eta^2_p = .08$) on visual dynamics (Fig 2B). Kruskal-Wallis testing confirmed the significant effect of frequency and rhythmicity on visual dynamics (H(3) = 76.3, $p < 0.001$; H(2) = 33.1, $p < 0.001$). Post-hoc Tukey tests revealed that 8 Hz, 10 Hz and 18 Hz stimulation elicited higher ratings of visual dynamics than 3 Hz (all $p < 0.001$). Moreover, ratings were higher for Rhythmic compared to Arrhythmic$_{norm}$ and Arrhythmic$_{pairs}$ ($p < 0.001$). For interaction effects, post-hoc Tukey tests showed that, during Rhythmic stimulation, 8 Hz, 10 Hz and 18 Hz elicited higher ratings of visual dynamics than 3 Hz ($p < .001$). Moreover, ratings were higher for Rhythmic compared to Arrhythmic$_{pairs}$ at 8 Hz, 10 Hz and 18 Hz (all $p < .001$) [See S1 Table for Tukey test results]. At 10Hz, visual dynamics were also higher for Rhythmic compared to Arrhythmic$_{norm}$ ($p = .03$) [See S1 Table for full report of Tukey tests]. Further, we found a significant

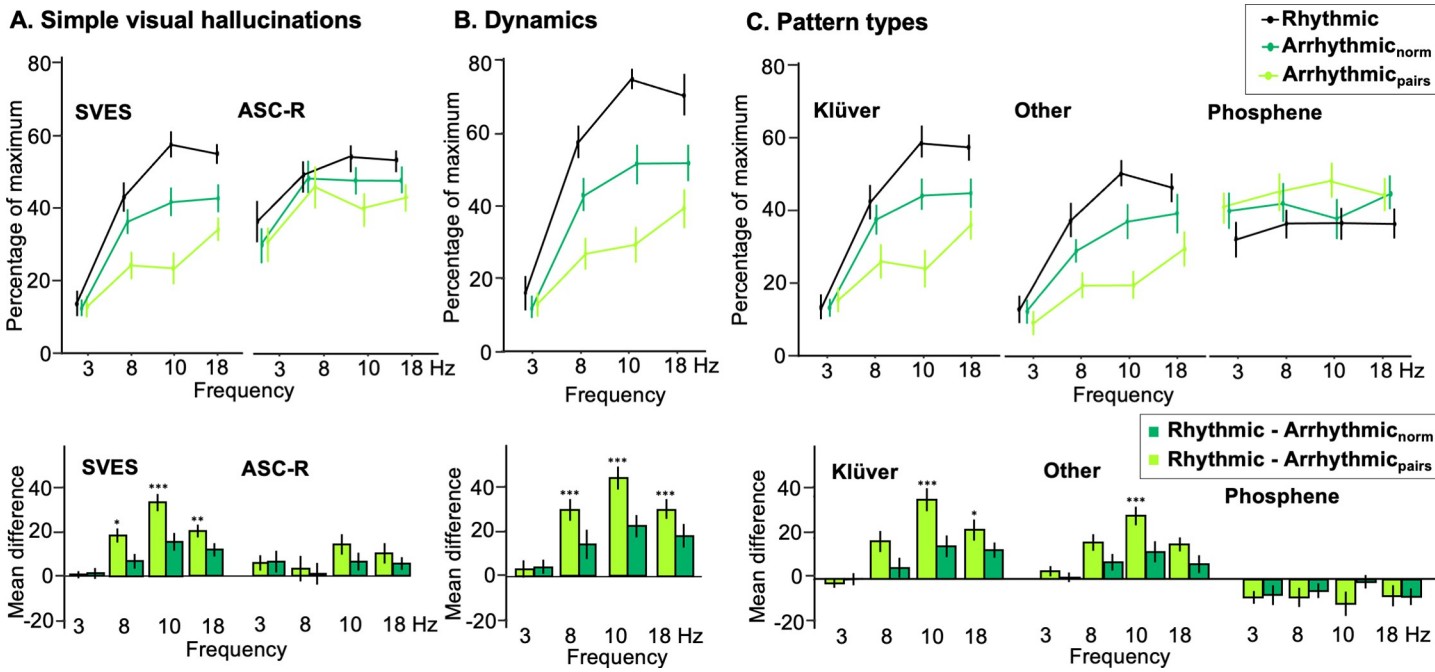

**Fig 2. Differential effects of rhythmicity and frequency on FLS-induced hallucinatory phenomena.** (A) Effects of rhythmicity and frequency on reports of simple visual hallucinations. Ratings are shown from the SVES *Simple Visual Hallucinations* and ASC-R *Elementary Imagery* scales. (B) Effects of rhythmicity and frequency on visual dynamics, which encompasses motion and how much the experience changes over time. (C) Effects of rhythmicity and frequency on different types of visual patterns. *Klüver forms* SVES subscale consists of spirals, cobwebs, targets and grids. *Other forms* include SVES items of rippling items and flowing lines. *Phosphene forms* refers to lower order forms and includes SVES items for TV snow and blobs of light or colour. Bar charts display the difference in ratings between arrhythmic controls and rhythmic stimulation for each frequency. Significance is determined by Tukey tests comparing ratings between rhythmicity conditions at each frequency level [See S1 Table for Tukey test results].

effect of rhythmicity ($F(2, 228) = 9.25$, $p < 0.001$, $\eta^2_p = .08$) and frequency ($F(3, 228) = 26.60$, $p < 0.001$, $\eta^2_p = .26$) on visual detail. Kruskal-Wallis testing confirmed the significant effect of frequency and rhythmicity on ratings of visual detail ($H(3) = 57.9$, $p < 0.001$; $H(2) = 13.5$, $p = 0.001$). Post-hoc Tukey tests revealed that visual detail was higher at 8 Hz, 10 Hz and 18 Hz compared to 3 Hz ($p < 0.001$) [See S1 Table for Tukey test results]. Ratings of visual detail were also higher for Rhythmic compared to Arrhythmic$_{pairs}$ ($p < 0.001$).

## Effects of frequency and rhythmicity on seeing different pattern types

To test if frequency and rhythmicity would affect the types of patterns experienced during simple visual hallucinations, we ran 3x4 ANOVAs on the *Klüver forms*, *Other forms*, *Phosphene forms* and *Paisley* SVES scales. We found that frequency had a main effect on ratings of seeing Klüver forms ($F(3, 228) = 34.92$, $p < 0.001$, $\eta^2_p = .31$) and Other forms ($F(3, 228) = 24.78$, $p < 0.001$, $\eta^2_p = .13$) (Fig 2C). Kruskal-Wallis testing confirmed significant effects of frequency on seeing Kluver forms ($H(3) = 69.6$, $p < 0.001$) and Other forms (($H(3) = 57.6$, $p < 0.001$). ANOVA testing also revealed a significant main effect of rhythmicity on Klüver form ($F(2, 228) = 17.47$, $p < 0.001$, $\eta^2_p = .13$) and Other form ratings ($F(2, 228) = 16.69$, $p < 0.001$, $\eta^2_p = .13$), which was further confirmed by Kruskal-Wallis testing (Kluver forms: $H(2) = 20.3$, $p < 0.001$; Other forms: $H(2) = 22.1$, $p < 0.001$). Additionally, there was a significant interaction effect on ratings of Klüver forms ($F(6, 228) = 3.26$, $p = 0.004$, $\eta^2_p = 0.08$). Post-hoc Tukey tests found that Klüver and Other forms generated higher ratings at 8 Hz, 10 Hz and 18 Hz than at 3 Hz (all $p < 0.001$). Ratings of Klüver and Other forms were also significantly higher

in the Rhythmic condition compared to arrhythmic controls [See S1 Table for Tukey test results]. While ANOVA testing identified a small effect of rhythmicity on ratings of Phosphene forms (F(2,228) = 3.07, $p$ = 0.05, $\eta^2_p$ = .03), Kruskal-Wallis testing found no effect of rhythmicity or frequency on Phosphene forms. Similarly, ANOVA testing identified a small effect of frequency (F(3, 228) = 2.71, $p$ = 0.05, $\eta^2_p$ = .03) and rhythmicity (F(2, 228) = 4.83, $p$ = 0.009, $\eta^2_p$ = .04) on perception of Paisley patterns, which was not supported by Kruskal-Wallis testing (i.e., no significant effects were found).

## Effect of rhythmicity and frequency on observed colours

Next, we explored whether there were categorical shifts in the spectrum of perceived colours during FLS at different frequencies and rhythmicity. No a priori hypotheses were set. The probability of each colour being selected is shown in Fig 3. From this, it appears there were no major shifts in the proportions of colours that were perceived across conditions. Descriptively, prominent peaks for reds, white and black can be observed, which increase in amplitude as the frequency increases. Logistic regression models were used to assess the relationship between frequency and rhythmicity and colour selection. The alpha threshold was Bonferroni corrected to 0.002 (0.05/24 due to 24 colours being tested). We found that frequency affected colour selection of white, light yellow and bright blue. The odds of participants selecting white were 6.0, 5.6 and 10.7 times greater during 8 Hz, 10 Hz and 18 Hz, respectively, compared to 3 Hz stimulation (all p < 0.001). White was chosen in 53.3% of trials at 3 Hz, 78.3% of trials at 8 Hz, 76.7% of trials at 10 Hz and 82.5% of trials at 18 Hz. Additionally, during 8 Hz stimulation, the odds of selecting light yellow were increased by 3.2-fold compared to 3 Hz stimulation (p < 0.001). At 18 Hz, the odds of selecting bright blue were 6.7 times higher than at 3 Hz (p < 0.001). Rhythmicity had a significant interaction with frequency on selection of white, whereby odds were 7.1 times higher during rhythmic stimulation at 10 Hz (p = 0.002).

## Effect of frequency and rhythmicity on complex visual hallucinations

Next, we tested the effects of frequency and rhythmicity on the occurrence of complex visual hallucinations. To this end, we performed a 3x4 ANOVA on the ratings of the SVES *Complex Imagery* scale. We found a main effect of rhythmicity (F(2, 228) = 4.35, p = 0.01, $\eta^2_p$ = .04), which was not found using nonparametric testing. However, nonparametric Kruskal-Wallis testing identified a significant effect of frequency on SVES complex imagery ratings (H(3) =

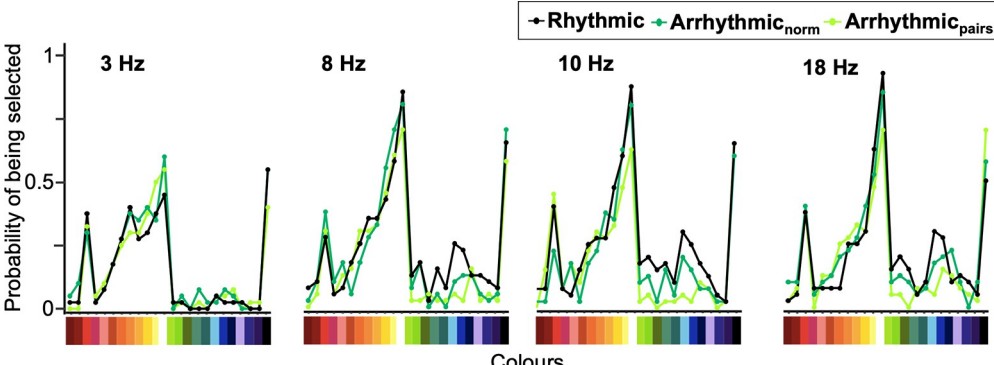

**Fig 3. Effects of rhythmicity and frequency on colour selection, depicted as a probability of each colour being selected.** This is expressed as sum of selections over the n = 20 participants. The mean selection probability is displayed, where for each participant the probability was based on the average of the first and second session (0, 0.5 or 1).

11.6, *p* = 0.009). When testing for complex hallucinations via the ASC-R *Complex Imagery* scale, no significant effect was found. Note that the ratings for complex visual hallucinations were overall relatively low, however showing a relevant variability (i.e., for SVES scale ratings: 3 Hz: 5.2 ± 12.4 *M* ± SD; 8 Hz: 10.3 ± 19.0; 10 Hz: 12.7 ± 19.4; 18 Hz: 11.1 ± 16.3), which motivates further exploration to identify factors that determine if a participant will experience complex hallucinations or not.

## Testing for a relationship between the personality trait absorptiveness and FLS-effects

In our sample, TAS scores, which indicate the personality trait Absorptiveness, ranged between 33 and 110 (maximum possible range: 0–136) across participants (*M* = 64.61, *SD* = 19.09). We used Pearson product-moment correlation to test if absorptiveness relates to the occurrence of simple and complex visual hallucinations. Across the four frequency conditions, no correlations (p<0.05) were found for SVES nor ASC-R ratings of simple visual hallucinations. Following the literature suggestion that the occurrence of complex hallucinations might be driven by a persons' absorptiveness, we tested within the four rhythmic frequency conditions for correlations of TAS with *Complex Imagery* scales of the SVES and ASC-R. We found a positive correlation for the 10 Hz condition, which was significant when assessed with the SVES (*r* = .61, *p* = .004) even after correction for multiple comparisons (Bonferroni: 0.05/ 8 = 0.00625). When testing with the ASC-R scores, this correlation was also present (*r* = .49, *p* = .03), however did not survive correction for multiple comparisons.

## Discussion

In this study, we aimed to determine the effects of flicker frequency and rhythmicity on FLS-induced hallucinatory effects. We used the well-established ASC-R in combination with an abridged version of the novel SVES to quantitatively assess the visual experience elicited by different FLS frequencies. This combination provided a direct comparison to previous data while also allowing a more thorough assessment of visual phenomena than previously attempted. We found effects of frequency on occurrence of simple visual hallucinations, especially perception of Klüver forms, as well as visual dynamics (e.g., motion) and the degree of visual detail. Here, participants reported that they experienced the most geometric patterns (e.g., Klüver forms) and visual dynamics during 10 Hz FLS. Furthermore, to test the influence of FLS rhythmicity on inducing simple visual hallucinations, we compared the subjective effects of rhythmic versus arrhythmic stimulation. We found that, even though arrhythmic stimulation delivered the same amount of physical light stimulation per second as rhythmic stimulation, it resulted in substantially reduced visual effects, including reduced perception of geometric patterns and visual dynamics. This may suggest that neural entrainment, elicited by rhythmic FLS, plays a significant role in the generation of simple visual hallucinations. The reduction in visual effects was most pronounced for the Arrhythmic$_{pairs}$ condition at 10 Hz, supporting its future use in investigations of the neuronal mechanisms underlying the flicker experience.

### Assessment of phenomenology

To draw conclusions from phenomenological data, it is first important to establish whether the employed assessment tools provide an accurate representation of the subjective experience. Based on recent work [1,2,17], we designed and administered an abridged version of the novel SVES to assess FLS-induced visual effects. We found that ratings of simple visual hallucinations were similar across ASC-R and SVES measures. Given that the ASC-R is well validated [22], the parity of these results indicates construct validity of the SVES, which should be

formally tested in future studies. Furthermore, larger effect sizes were found using SVES ratings compared to ASC-R ratings. This is likely because the SVES was designed to capture specific details and pattern subtypes within simple visual hallucinations while ASC-R targets an overarching altered state experience. By differentiating pattern subtypes, we found that some visual phenomena occur irrespective of frequency and rhythmicity (i.e., phosphene forms; see Fig 2C), thus the SVES preserves this information while the ASC-R only captures gross visual phenomena: "I saw patterns". Indeed, recent studies did not find differences in ASC-R ratings of simple visual hallucinations at different FLS frequencies [2,3], even though the experience intensity was rated differently [3]. While the ASC-R questionnaire remains useful for comparisons across altered state induction methods [2,8], the SVES enables a more detailed assessment of visual hallucinatory characteristics.

## Effects of frequency and rhythmicity on visual hallucinatory phenomena

First, we tested how the phenomenal characteristics of FLS-induced hallucinatory phenomena were affected by flicker frequency. We found that simple visual hallucinations, such as perception of Klüver forms, were experienced most intensely at 10 Hz rhythmic FLS. This not only confirms previous findings, where 10 Hz FLS was identified as generating the greatest hallucinatory effects [2,3,17,28], but offers an extension by distinguishing phenomenal components within the experience. For example, by differentiating simple visual hallucinations into patterns subtypes, we found that Klüver forms (i.e., grids, cobwebs, spirals, tunnels) were the most reported pattern subtype during rhythmic FLS at 8 Hz, 10 Hz and 18 Hz. Furthermore, we found that 10 Hz rhythmic stimulation elicited the most visually dynamic experiences. Visual dynamics encompass perceived motion and how much the experience changes over time. FLS-induced moving patterns have been previously documented [1,31,32]. We extend this by finding that flicker frequency had the largest effect on dynamics compared to all other FLS-induced subjective qualities, emphasising that it constitutes a highly relevant characteristic of FLS effects. Future studies could incorporate eye tracking sensors that monitor participant eye movements during the flicker experience to explore whether participants' eye position and movement adds to variability in the subjective experience. Altogether, out of the tested frequencies, our results identify 10 Hz FLS as the frequency that induced the greatest perceptual changes.

Next, we investigated whether rhythmicity affected the phenomenal characteristics induced by FLS. We found that arrhythmicity significantly reduced simple visual hallucinations and visual dynamics. The relative reduction of effects was largest for Arrhythmic$_{pairs}$ at 10 Hz, compared to rhythmic 10 Hz. The use of paired flashes as a control condition was first utilised by Ffytche [28], where it was found that paired flashes led to significant decreases in occipito-temporal activity, measured via EEG, compared to periodic FLS. However, it should be noted that Ffytche did not include a phenomenal characterisation of the flicker conditions. Furthermore, we applied an arrhythmic version of the paired flash stimulation whereby inter-pair intervals were sampled from an exponential probability distribution. In case of higher frequencies (i.e., 18 Hz), flicker trains of higher frequency due to IFI randomization were more likely to occur than in the other conditions, with local frequency of up to 37 Hz (compare Fig 1). Due to the randomization of IFIs within one second, these trains were very short and interrupted by longer IFIs, making it unlikely that they were majorly driving the subjective experiences. Overall, we present considerable evidence to show that frequency-matched arrhythmic FLS reduces hallucinatory effects compared to rhythmic stimulation, underscoring the importance of rhythmicity in determining the intensity of FLS effects.

Simple visual patterns, such as those reported in our study, are also commonly reported following administration of a range of psychedelic drugs (serotonin-2A receptor agonists) [19,33]. Indeed, Klüver forms were first identified in the context of mescaline-induced hallucinations [15] and have also been reported in migraine aura [5] and Charles Bonnet Syndrome (i.e., reported mosaic patterns as a form of grid) [34]. Similarly, perceived motion of visual imagery also occurs in epileptic seizures [35] and during migraine aura [4]. The similarities in reports of simple visual hallucinations across aetiologically distinct origins (including FLS) indicate shared underlying neural mechanisms. Seminal computational modelling work suggests that the structure of simple visual hallucinations is to some extent determined by the neuronal architecture between the retina and brain [14,16,36,37]. Therefore, simple visual hallucinations may reveal the hidden architecture of visual areas of the brain. Future neurophysiology research can therefore draw upon research from various domains to formulate a better understanding of how hallucinatory phenomena are generated.

Aside from simple hallucinatory phenomena, there were small effects of rhythmicity and frequency on reports of complex visual hallucinations. Complex hallucinations involve the perception of realistic objects, scenes, and faces (i.e., containing semantic value). While previous reports found them to be more prevalent at 3 Hz stimulation [3], we found that ratings increased with frequency. Still, complex visual hallucinations remained relatively low throughout all FLS conditions and occurred to a lesser extent than simple hallucinations, which is in line with previous work [2]. This reinstates that FLS reliably induces simple visual hallucinations, while phenomena that involve semantically meaningful content occur only occasionally.

To explore what factors may influence the extent of experiencing complex hallucinations, we tested their occurrence in relation to the personality trait "absorptiveness", following from previous work [2]. We found a positive correlation between absorptiveness and complex imagery ratings for 10 Hz rhythmic FLS. Absorptiveness positively correlates with hypnotisability [21,38], a term that precedes the recently introduced concept of "phenomenological control" [39], which describes one's capacity to alter their subjective experience in order to meet expectations. This could suggest that participants with high absorptiveness experience more hallucinations due to expectation that they will occur. Interestingly, however, absorptiveness did not correlate with simple visual hallucinations. Following from this, predictive coding models suggest that altered hierarchical processing, more specifically prior distributions, on either lower or higher levels of the visual hierarchy relate to simple or complex hallucinations, respectively [40–42]. In light of this distinction, it is plausible that personality traits like absorptiveness and phenomenological control influence hierarchically higher regions and thereby increase likelihood of experiencing complex hallucinations. Further research could expand the scope of assessing how inter-individual differences influence FLS-induced phenomenology and neural processing by measuring a wider variety of participant traits, such as phenomenological control.

## Exploration of FLS-induced perception of colours

We explored the types of colours that participants reported during different levels of flicker frequency and rhythmicity. We aimed to decipher whether there were categorical shifts in the spectrum of perceived colours depending on the type of FLS. We found that increasing flicker frequency increased the chances of participants reporting the perception of white, light yellow and bright blue. Rhythmicity also influenced the selection of white, whereby rhythmic stimulation led to higher chances of perceiving white during 10 Hz stimulation. Previous research identified that colours are often experienced during rhythmic FLS [2,3], Ganzfeld stimulation [43], psychedelic drug-induced experiences [19,33], epileptic seizures [6] and Charles Bonnet

Syndrome [34], however we are not aware of literature that has further classified the specific colours experienced or their respective proportions within an experience. The VES enables group-level quantification of each colour perceived during different FLS conditions. It is interesting to note that colour perception is only weakly modulated by frequency and rhythmicity, especially when compared against simple patterns and visual dynamics. This could indicate that hallucinatory colour perception arises in the lower levels of the visual pathway (e.g., from retinal stimulation; entoptic phenomena), while patterns and other hallucinatory phenomena depend on frequency and rhythmicity-dependent neural mechanisms, such as neural entrainment. These exploratory findings can be used to formulate hypotheses of flicker-induced colour perception in future studies.

## Potential underlying neural mechanisms

Ultimately, it is of interest to explain phenomenal characteristics in relation to their underlying neural mechanisms. While there have been some neuroimaging studies of FLS [3,26–28], the direct link between FLS-induced phenomena and neural activity is yet to be established. Currently, there are three main views addressing the neural mechanisms that lead to FLS-induced visual hallucinatory phenomena.

Firstly, it is likely that neural entrainment plays an important role. Haegens [44] defined entrainment as the phase alignment of existing brain oscillations to an external periodic stimulus, which continues for several cycles after stimulus termination. Previous EEG studies found that rhythmic flicker at alpha frequency increases neural entrainment at that frequency [3,12,45–47]. Further, it was found that rhythmic flicker produced stronger phase locking than arrhythmic stimulation when presented with high light intensity at a stimulation frequency close to the individual's dominant intrinsic frequency [45–47]. In our study, we found the greatest differences in reported hallucinatory phenomena between Rhythmic and the Arrhythmic$_{pairs}$ control, which has more arrhythmicity than the normally distributed control, as determined by the RMSSD of IFIs (see Methods). This finding could suggest that the relationship between rhythmicity and visual effects exists as a continuum where the degree of arrhythmicity affects the extent to which effects are reduced. However, it is important to note that we did not directly assess the neural effects of arrhythmic stimulation. In this light, it should be considered that other mechanisms can also contribute to oscillatory activity, such as the superposition of event-related responses, which are evoked cortical responses to visual stimulation that add onto, but do not interact with, ongoing oscillations [48]. Moreover, a recent study using rhythmic flickering checkerboards found evidence for both frequency-specific neural responses, supporting the entrainment model, and frequency-independent resonance phenomena, supporting the superposition model [49]. As we found that intensity of FLS effects was affected by frequency, it is likely that underlying frequency-specific neural responses, such as entrainment, contribute to FLS effects. However, future EEG research is necessary to test whether FLS-induced neural responses satisfy the criteria for entrainment [44] and if there is markedly less entrainment elicited by arrhythmic conditions. In doing so, evidence can be provided to determine whether neural entrainment mediates the effect of frequency and rhythmicity on the flicker-induced subjective experience.

Secondly, the Ermentrout-Cowan model proposes that the perception of Klüver forms corresponds to self-organised striped cortical activity in the primary visual cortex (V1) [14,16,17,50]. Due to the nonlinear transformation of retinal to cortical coordinates, the model demonstrates that striped activation in V1 translates into spirals, tunnels and other Klüver forms when mapped onto retinal coordinates. The model incorporates anatomical knowledge of the visual cortex, such as the size of V1 hypercolumns, their lateral inhibitory connections

and orientation selectivity [14,16]. Nevertheless, it cannot explain the entirety of reported simple visual hallucinations as there are other pattern types and characteristics that it does not account for. For example, Ffytche [28] found that FLS led to increased V4 activity, which may correspond to perceived colour or motion of patterns. Still, our findings lend some support to the model as we found that Klüver forms are reported to a greater degree than other patterns subtypes, reinforcing their relevance in the study of simple visual hallucinations. Moreover, as expected, flicker frequency and rhythmicity did not influence whether phosphene forms were perceived. This supports the notion that phosphene forms, such as TV snow and blobs, are generated from retinal stimulation with a bright light source [17] as all frequency and rhythmicity levels produced the same amount of light input (300ms of light stimulation per second). The additional patterns, such as Klüver forms, are more likely to be a result of frequency-dependent modulation of neural activity via periodic light stimulation.

Finally, one can look to models developed in other domains to inform predictions of how FLS-induced phenomena could arise. The cortico-striato-thalamo-cortical (CSTC) model proposes that drug- and pathology-induced hallucinations are associated with aberrant modulation of thalamus activity leading to thalamocortical dysconnectivity [51,52]. This is supported by studies that found drug-induced alterations in sensory perception to be positively correlated with the functional connectivity between thalamus and primary sensory cortices [53]. Such increased thalamocortical functional connectivity has also been found during flicker-induced hallucinations [28] and psychosis [54–56]. Further, thalamocortical dysconnectivity has been implicated in Ganzfeld-induced altered states [57] and thalamocortical functional and structural dysconnectivity is present in patients with epilepsy [58–60] and migraine [61–63]. These findings hint that thalamocortical dysconnectivity may also play a key role in FLS-induced effects.

It is likely that the three views are not mutually exclusive but that their proposed mechanisms interact or influence each other at different levels. For example, thalamocortical dysconnectivity may arise from neural entrainment at specific frequencies. Furthermore, the Ermentrout-Cowan model may explain specific properties of the visual experience, such as perception of Klüver forms, but requires additional inputs from other models to encapsulate the entire phenomenal experience. The aim of future research should be to tie together the interacting mechanisms in order to formulate an overarching model of how simple visual hallucinations are generated in the brain.

## Outlook

Here, we have presented the effects of flicker light rhythmicity and frequency on aspects of the flicker-induced experience, such as simple visual hallucinations, visual dynamics and perceived colours. The applied SVES generated similar ratings to the ASC-R, but with larger effect sizes, which suggests that SVES can capture the FLS-induced phenomenology with a higher level of detail. Further, we found that flicker arrhythmicity significantly reduced visual effects, which implies that neural entrainment may be critical to the generation of simple visual hallucinations. From two frequency-matched arrhythmic control conditions, we identified the Arrhythmic$_{pairs}$ condition as most effective in reducing simple visual hallucinations when compared against rhythmic stimulation, especially at 10 Hz. Using this, future neuroimaging studies can investigate the neural mechanisms that mediate the effects of rhythmicity on the flicker-induced hallucinatory experience.

## Supporting information

**S1 Table. Tables for full report of Tukey test results.**
(PDF)

**S1 Appendix. Abridged version of Stroboscopic Visual Experience Survey.**
(PDF)

## Acknowledgments

We would like to thank Tom Galea, Jay Conlon and Lumenate Growth Inc. for the helpful discussions and generous provision of experimental hardware.

## Author Contributions

**Conceptualization:** Ioanna Alicia Amaya, David John Schwartzman, Timo Torsten Schmidt.

**Data curation:** Ioanna Alicia Amaya.

**Formal analysis:** Ioanna Alicia Amaya, Nele Behrens, Timo Torsten Schmidt.

**Funding acquisition:** Timo Torsten Schmidt.

**Investigation:** Ioanna Alicia Amaya.

**Methodology:** Ioanna Alicia Amaya, David John Schwartzman, Trevor Hewitt.

**Project administration:** Timo Torsten Schmidt.

**Resources:** Timo Torsten Schmidt.

**Software:** Ioanna Alicia Amaya.

**Supervision:** Timo Torsten Schmidt.

**Visualization:** Ioanna Alicia Amaya, Timo Torsten Schmidt.

**Writing – original draft:** Ioanna Alicia Amaya.

**Writing – review & editing:** David John Schwartzman, Trevor Hewitt, Timo Torsten Schmidt.

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
