## [Decision Letter · Decision Letter 0]

26 Jan 2023

PONE-D-22-33362Effect of frequency and rhythmicity on flicker light-induced hallucinatory phenomenaPLOS ONE

Dear Dr. Schmidt,

Thank you for submitting your manuscript to PLOS ONE. After careful consideration, we feel that it has merit but does not fully meet PLOS ONE’s publication criteria as it currently stands. Therefore, we invite you to submit a revised version of the manuscript that addresses the points raised during the review process.

Please address all reviewers' comments.

We look forward to receiving your revised manuscript.

Kind regards,

Manuel Spitschan

Academic Editor

PLOS ONE

Journal Requirements:

"The investigator-initiated study was financially supported by a donation from Lumenate Growth Ltd to TTS."

"I have read the journal's policy and the authors of this manuscript have

the following competing interests:

TTS: This research was supported by a unrestricted donation from Lumenate

Growth ltd to Freie Universität Berlin allocated to TTS."

We note that you received funding from a commercial source: Lumenate Growth ltd 

Within this Competing Interests Statement, please confirm that this does not alter your adherence to all PLOS ONE policies on sharing data and materials by including the following statement: ""This does not alter our adherence to PLOS ONE policies on sharing data and materials.” (as detailed online in our guide for authors http://journals.plos.org/plosone/s/competing-interests).  If there are restrictions on sharing of data and/or materials, please state these. Please note that we cannot proceed with consideration of your article until this information has been declared. 

5. We note that Figures 1 and S1 Appendix in your submission contain copyrighted images. All PLOS content is published under the Creative Commons Attribution License (CC BY 4.0), which means that the manuscript, images, and Supporting Information files will be freely available online, and any third party is permitted to access, download, copy, distribute, and use these materials in any way, even commercially, with proper attribution. For more information, see our copyright guidelines: http://journals.plos.org/plosone/s/licenses-and-copyright.

a. You may seek permission from the original copyright holder of Figures 1 and S1 Appendix to publish the content specifically under the CC BY 4.0 license. 

6. We note that Figure S1 Appendix includes an image of a participant in the study. 

Reviewers' comments:

Reviewer's Responses to Questions

**Comments to the Author**

1. Is the manuscript technically sound, and do the data support the conclusions?

Reviewer #1: Yes

Reviewer #2: Yes

2. Has the statistical analysis been performed appropriately and rigorously? 

Reviewer #1: Yes

Reviewer #2: Yes

3. Have the authors made all data underlying the findings in their manuscript fully available?

Reviewer #1: Yes

Reviewer #2: Yes

4. Is the manuscript presented in an intelligible fashion and written in standard English?

Reviewer #1: Yes

Reviewer #2: Yes

5. Review Comments to the Author

Reviewer #1: Review of “Effect of frequency and rhythmicity on flicker light-induced hallucinatory phenomena”.

The authors report an experiment investigating the frequency and rhythmicity of flickering light on simple visual hallucinations. They found the experience of visual hallucinations to be maximal at 10 Hz and that rhythmic light resulted in stronger hallucinations than an arrhythmic control condition.

Overall I find no major flaws in the study. This is a very interesting topic and an important step towards understanding the phenomena.

The following should be considered minor revisions/suggestions.

Some additional information on the experimental setup should be added to the methods section. There were 12 LEDs, presumably these were white LEDs? This should be stated for clarity.

What pattern were the LEDs positioned in, e.g. in a circle, a grid? How far apart?

Importantly, the visual angle of the light source should be reported because peripheral flickering light is perceived differently from flickering light on the fovea. Although the light is diffuse because the eyes are closed, the visual angle of the light source could be a factor on the resulting perceptions. This can simply be calculated from the distance and width of the light source.

A potential factor which should be considered is the position of the eyes when eyelids are closed. In my experience participants sometimes report moving their eyes to look away from a bright light sources with eyes closed, as this reduces the unpleasantness of bright light. Furthermore, it is known that it is difficult to maintain a consistent position of the eye with eyes closed, so the position might change over time. This is not a critical problem for the current study because there is no reason to suspect that the eye position was different across control conditions, but it should be mentioned in the discussion as a potential source of variability in the subjective experience.

A second control condition using paired arrhythmic stimulation was used. It is not clear why an additional control condition is needed, when the arrhythmic condition controls for rhythmicity. Perhaps the authors could explain the decision to add an extra control condition in more detail in the introduction. For example, do the authors hypothesize that the decreases in occipitotemporal EEG activity found by Ffytche is not present in the normal arrhythmic control condition? The authors report that the paired arrhythmic stimulation can be considered more arrhythmic, it would be good if they could elaborate on why this is the case. Is it simply a greater variation in inter flash intervals? Do the authors have any hypothesis as to why the reduction in visual effects was most pronounced for the arrhythmic pairs condition?

The authors discuss the possibility of neural entrainment being responsible for the generation of simple visual hallucinations. While I agree that neural entrainment is a possible (even likely) explanation, there is a debate in the EEG literature as to whether the frequency specific neural responses to visual flicker (SSVEPs) are the result of entrainment of ongoing oscillations, or a resonance phenomena where the cortex has a preferred frequency of stimulation. For example Capilla et. al. (2011) argued SSVEPs can be explained as a superposition of transient responses to the visual stimulation. In a recent study we have shown that different cortical sources of alpha oscillations show very different steady state responses to visual flicker, which vary depending on the distance of the flicker frequency and the individual’s alpha frequency (Nuttall et. al. 2022). In the current study the authors have shown that rhythmicity is important for the hallucinatory visual effect, so an interaction between neural oscillations and the flicker is likely, but entrainment is only one possible explanation. Distinguishing between entrainment and resonance is difficult and the exact definition of entrainment is not always clearly defined, see Haegens (2020) for a good discussion. It is quite possible that different brain areas respond to rhythmic stimulation in a variety of ways, with some showing entrainment of an ongoing oscillation, and some simply responding preferentially to an optimal frequency of rhythmic input. I appreciate that a full discussion of these issues is beyond the scope of the current article, but it would be good to acknowledge the possibility of different neural mechanisms which might explain this effect.

James Dowsett

References:

Capilla A, Pazo-Alvarez P, Darriba A, Campo P, Gross J (2011) Steady-State Visual Evoked Potentials Can Be Explained by Temporal Superposition of Transient Event-Related Responses. PLoS ONE 6(1): e14543.

Nuttall, R., Jäger, C., Zimmermann, J., Archila-Melendez, M. E., Preibisch, C., Taylor, P., ... & Dowsett, J. (2022). Evoked responses to rhythmic visual stimulation vary across sources of intrinsic alpha activity in humans. Scientific reports

Saskia Haegens (2020): Entrainment revisited: a commentary on Meyer, Sun,

and Martin (2020), Language, Cognition and Neuroscience, DOI: 10.1080/23273798.2020.1758335

Reviewer #2: This paper examines – as summarised in the title – the effect of frequency and rhythmicity on various flicker-induced visual hallucinations. As described in the paper, 10-Hz visual stimulation has been reported to lead to strongest hallucinations (as compared to other stimulation rates), but the effect of rhythmicity remained unclear due to the lack of well-designed arrhythmic control conditions. Such conditions, along with novel questionnaires, were used in the current study. The authors were able to confirm the advantage of 10-Hz flicker (among several other rates) to induce visual hallucinations. In addition, they show that, in particular for 10 Hz, rhythmic stimulation leads to stronger hallucinatory effects than arrhythmic control stimulation.

This paper is very well written and the rationale is easy to follow. I cannot see any major issues with this work and believe that this is a well-designed and executed study. I do have two minor comments:

- I was a bit confused about arrhythmic pairs control condition. The rate (frequency) of paired flashes always seems faster than in the corresponding rhythmic condition. Given the effect of rate on perception, I wonder if this can lead to an apparent effect of rhythmicity that is due to this difference in “local” frequency/rate.

- For some statistical tests, the authors report a reliable interaction of rhythmicity and frequency. In some cases (e.g., page 13), it remains unclear which combination of conditions has produced the interaction. This could be described in more detail, in particular as main effects can be difficult to interpret in the presence of such an interaction.

6. PLOS authors have the option to publish the peer review history of their article (what does this mean?). If published, this will include your full peer review and any attached files.

Reviewer #1: **Yes: **James Dowsett

Reviewer #2: **Yes: **Benedikt Zoefel

---

## [Author Response · Author response to Decision Letter 0]

9 Mar 2023

Reviewer 1: We thank you for the positive reception of our manuscript and for the helpful comments, suggestions, and additional literature. We have incorporated all your comments to the revised manuscript. 

Reviewer 2: We thank you for the positive feedback and constructive comments for improving our manuscript. We have incorporated all your points to the revised manuscript.

---

## [Decision Letter · Decision Letter 1]

29 Mar 2023

Effect of frequency and rhythmicity on flicker light-induced hallucinatory phenomena

PONE-D-22-33362R1

Dear Dr. Schmidt,

We’re pleased to inform you that your manuscript has been judged scientifically suitable for publication and will be formally accepted for publication once it meets all outstanding technical requirements.

Kind regards,

Manuel Spitschan

Academic Editor

PLOS ONE

Additional Editor Comments (optional):

Reviewers' comments:

Reviewer's Responses to Questions

**Comments to the Author**

1. If the authors have adequately addressed your comments raised in a previous round of review and you feel that this manuscript is now acceptable for publication, you may indicate that here to bypass the “Comments to the Author” section, enter your conflict of interest statement in the “Confidential to Editor” section, and submit your "Accept" recommendation.

Reviewer #1: All comments have been addressed

Reviewer #2: All comments have been addressed

2. Is the manuscript technically sound, and do the data support the conclusions?

Reviewer #1: Yes

Reviewer #2: Yes

3. Has the statistical analysis been performed appropriately and rigorously? 

Reviewer #1: Yes

Reviewer #2: Yes

4. Have the authors made all data underlying the findings in their manuscript fully available?

Reviewer #1: Yes

Reviewer #2: Yes

5. Is the manuscript presented in an intelligible fashion and written in standard English?

Reviewer #1: Yes

Reviewer #2: Yes

6. Review Comments to the Author

Reviewer #1: (No Response)

Reviewer #2: (No Response)

7. PLOS authors have the option to publish the peer review history of their article (what does this mean?). If published, this will include your full peer review and any attached files.

Reviewer #1: **Yes: **James Dowsett

Reviewer #2: **Yes: **Benedikt Zoefel

---

## [Editor Report · Acceptance letter]

3 Apr 2023

PONE-D-22-33362R1 

Effect of frequency and rhythmicity on flicker light-induced hallucinatory phenomena 

Dear Dr. Schmidt:

I'm pleased to inform you that your manuscript has been deemed suitable for publication in PLOS ONE. Congratulations! Your manuscript is now with our production department. 

Kind regards, 

on behalf of

Dr. Manuel Spitschan 

Academic Editor

PLOS ONE